# Inflammasome in Skeletal Muscle: NLRP3 Is an Inflammatory Cell Stress Component in Inclusion Body Myositis

**DOI:** 10.3390/ijms241310675

**Published:** 2023-06-26

**Authors:** Karsten Kummer, Imke Bertram, Sabrina Zechel, Daniel B. Hoffmann, Jens Schmidt

**Affiliations:** 1Department of Neurology and Pain Treatment, Neuromuscular Center, Center for Translational Medicine, Immanuel Klinik Rüdersdorf, University Hospital of the Brandenburg Medical School, 15562 Rüdersdorf bei Berlin, Germany; 2Faculty of Health Sciences Brandenburg, Brandenburg Medical School, 15562 Rüdersdorf bei Berlin, Germany; 3Department of Neurology, Neuromuscular Center, University Medical Center Göttingen, 37075 Göttingen, Germany; 4Department of Neuropathology, University Medical Center Göttingen, 37075 Göttingen, Germany; 5Department of Trauma, Orthopaedic and Plastic Surgery, University Medical Center Göttingen, 37075 Göttingen, Germany

**Keywords:** NLRP3, inflammasome, inclusion body myositis, muscle inflammation, NLRC5

## Abstract

Inclusion body myositis (IBM) is a chronic, mostly treatment-resistant, inflammatory myopathy with a pathology that centers around specific interactions between inflammation and protein accumulation. The study aimed to identify the inflammasome as a key event in the complex network of pathomechanisms. Regulation of the inflammasome was assessed in a well-established pro-inflammatory cell culture model using human myoblasts and primary human myotubes. By quantitative PCR, western blot and immunocytochemistry, inflammasome markers including NLRP3 were assessed in muscle cells exposed to the cytokines IL-1β and IFN-γ. The data were corroborated by analysis of muscle biopsies from patients with IBM compared to other myositis subtypes. In the cell culture model of IBM, the NLRP3 inflammasome was significantly overexpressed, as evidenced by western blot (*p* = 0.03) and quantitative PCR (*p* < 0.01). Target genes that play a role in inflammasome assembly, T-cell migration, and MHC-I expression (*p* = 0.009) were highly co-upregulated. NLRP3 was significantly overexpressed in muscle biopsies from IBM samples compared to disease controls (*p* = 0.049), including other inflammatory myopathies. Due to the extraordinary features of the pathogenesis and the pronounced upregulation of NLRP3 in IBM, the inflammasome could serve as a key molecule that drives the inflammatory cascade as well as protein accumulation in the muscle. These data can be useful for future therapeutic developments.

## 1. Introduction

Inclusion body myositis (IBM) belongs to the group of idiopathic inflammatory myositis (IIM) [1]. For the total population, an incidence of 4.9–14.9 cases per 1 million is given. However, in the age group over 50 years, an incidence of 35.3/1,000,000 has been demonstrated [2,3]. IBM is characterized by a progressive course with increasing asymmetric muscle weakness of the distal and proximal skeletal muscles [4]. 

The pathogenesis of IBM is still uncertain. However, the histological findings show a combination of inflammatory and degenerative mechanisms [1]. 

Muscle fibers of IBM patients show an upregulation of the antigen-presenting major histocompatibility complex I (MHC-I) [5]. Thus, muscle fibers appear to take over the function of professional antigen-presenting cells (APCs) and cause clonal expansion and activation of CD8^+^ T cells in their environment [6,7]. Activated endomysial CD8^+^ T cells have been demonstrated to form an immunological synapse with antigen-presenting myofibers via non-classical co-stimulatory molecules [5,6].

In addition, it has been demonstrated that muscle fibers themselves create a pro-inflammatory milieu by releasing pro-inflammatory cytokines (IFN-γ, TNF-α, IL-1β and IL-6). This results in further migration of T cells and retention of inflammation [8,9]. The inflammatory activity seems to correlate with the extent of the degenerative changes which in particular are represented by high levels of interleukin 1-β (IL-1β) and associated with the accumulation of the amyloid precursor protein (APP) [10,11].

Histopathological manifestations of degeneration are present in the form of protein aggregates, i.e., inclusion bodies. Inclusion bodies are mainly composed of β-amyloid [10], phosphorylated tau protein, presenilin, p62, and α-synuclein. Furthermore, the autophagic activity in IBM patients appears to be impaired, which leads to further accumulation of degenerative proteins in the fibers [11,12].

The NLRP3 (nucleotide-binding oligomerization domain (NOD)-like receptor family pyrin domain containing 3) inflammasome is a macromolecular complex, activated by a variety of endogenous and exogenous stimuli. The activated complex triggers the release of pro-inflammatory IL-1β and IL-18 via ASC and activation of pro-caspase 1 [13]. Previous studies revealed an upregulation of the NLRP3 inflammasome in age-related neurodegenerative disorders such as Parkinson’s disease [14] and Alzheimer’s disease [15], but also in Duchenne muscular dystrophy [16]. Because intracellular protein accumulation and endogenous cell stress have been shown to be activators of pro-inflammatory protein complexes it is reasonable to conclude that inflammasomes are a further link between inflammation and degeneration [15,17].

Here, we identify a strong overexpression of NLRP3 inflammasome as a cell stress marker in the skeletal muscle of IBM patients and a well-established muscle cell culture model of the disease.

## 2. Results

### 2.1. IBM Muscle Tissue Shows an Upregulation of NLRP3

For this ex vivo study, we used patient-derived muscle biopsy material. Biopsied patients without inflammatory changes in histological evaluation served as the control group. Comparative analyses were performed on patients with the diagnosis of IBM, dermatomyositis (DM), polymyositis (PM), necrotizing myopathy (NM) and with hereditary muscular dystrophies. Samples from a total of 68 patients were examined, as detailed in Table 1. A comparable number of samples from male and female patients were included.

Significantly increased NLRP3-mRNA levels were observed in samples from IBM compared to controls. For PM, DM, NM and dystrophies, no increase of NLRP3 was noted (Figure 1A).

The mRNA level of NLRC5, a key activator of MHC class I transcription [18], was significantly overexpressed in IBM samples compared to controls (Figure 1B). In the myositis subtypes DM and PM, and in muscular dystrophies, NLRC5 was somewhat upregulated, but this did not reach statistical significance.

By immunohistochemistry, an elevated expression level of NLRP3 was observed in IBM muscle tissue in comparison to the control muscle (Figure 2).

### 2.2. NLRP3 Inflammasome Is Upregulated under Pro-Inflammatory Cell Stress In Vitro

Potential effects of the inflammatory milieu on protein level in myotubes were investigated using an indirect detection via an immunofluorescence assay. Prior to the immunostaining procedure, the primary myotube cultures were treated with IL-1β and IFN-γ for 24 h, 48 h, and 72 h, respectively. The fluorescence signal of NLRP3 in unstimulated cells, and upon IL-1β and IFN-γ treatment, is shown in Figure 3A. MHC-I staining was performed as a positive control for an efficient pro-inflammatory stimulation (Figure 3A). For both NLRP3 and MHCI, an increased signal intensity was detected. Upregulation of NLRP3 was evidenced by an automated, unbiased grey-scale analysis of the photomicrographs. The signal intensity for each individual channel was assessed. After background subtraction, mean intensities were calculated for each individual sample (Figure 3B). A significant upregulation of the MHC-I complex upon stimulation with IL-1β and IFN-γ was shown for all time points (24 h, 48 h, 72 h). There was a tendency for signal increase of NLRP3 upon stimulation compared to the untreated control, which did not reach statistical significance (Figure 3B). 

We determined the protein expression of NLRP3 by western blot analysis of cell lysates from untreated and stimulated myotube cultures (Figure 3C). The signal intensity of the NLRP3-positive band relative to the endogenous control β-actin was analyzed using Image J software (Figure 3D). The statistical analysis revealed a significant difference of expression levels of NLRP3 in stimulated cells compared to the untreated control cells.

### 2.3. Expression of NLRP3 Increases In Vitro in Muscle Cells in Pro-Inflammatory Conditions 

We determined the mRNA level of the inflammasome protein NLRP3 using quantitative (realtime) PCR (qPCR) in muscle cells stimulated with IL-1β and IFN-γ for 24 h and untreated control cells. We found a significant increase of the mRNA level for NLRP3 in stimulated cells compared to the untreated control cells (Figure 3E). 

### 2.4. NLRP3 Expression Correlates with Pro-Inflammatory Markers in Muscle Biopsies from IBM Patients

The expression of NLRP3 was correlated with different markers for cell stress and myodegeneration for a subset of the IBM muscle samples.

The mRNA expression of NLRP3 correlated significantly with the tagging molecule ubiquitin (Figure 4). Moreover, NLRP3 expression was significantly associated with the pro-inflammatory molecules TNF-α and TGF-β. The expression levels of APP somewhat correlated to the expression of NLRP3 (Figure 4), but this did not reach statistical significance.

## 3. Discussion

IBM is a relentlessly progressive, acquired myositis of middle-aged individuals. The exact pathogenesis of IBM has not yet been clarified, but degenerative and inflammatory mechanisms appear to be similarly involved during initiation and perpetuation of the disease. Links between these two pathways indicate a mutual interplay of mechanisms. Among these, IL-1β and NO-stress, as well as dysfunctional autophagic processing, have been demonstrated [1,10]. 

The NLRP3 inflammasome is an important catalyst for the release of IL-1β in the innate immune response. Many overlaps between the previously known activators of this inflammasome and components of IBM protein aggregations suggest that the NLRP3 inflammasome is a central component of the interplay between inflammation and degeneration in IBM muscle and its model systems. In previous observations, it has been shown that β-amyloid was able to induce NLRP3 inflammasome activation [19] and, conversely, NLRP3 activation led to a profound β-amyloid-pathology in an in vivo Alzheimer’s model [15]. Most recently, the same effect could be demonstrated for tau-pathology [20]. In skeletal muscle, an upregulation of NLRP3 was shown in Duchenne muscular dystrophy [16] and in Valosin-Containing Protein (VCP) Myopathy [21]. NLRP3 knock-out in mdx mice or blockade of NLRP3 inflammasome in VCP mice led to a reduction in inflammation and oxidative stress in dystrophic muscle and resulted in a higher global muscle force. 

Our results show an upregulation of NLRP3 in the well-established in vitro model for chronic muscle inflammation, as used for IBM [11] upon stimulation with IL-1β and IFN-γ for 24 h, 48 h, and 72 h, creating a proinflammatory milieu. This in vitro model has been used extensively in previous studies to evaluate the muscle cell response to pro-inflammatory stress signals in IBM muscle samples and cell culture [11,22], e.g., autophagy [23] and accumulation of β-amyloid [24]. Moreover, we could demonstrate an upregulation of NLRP3 in samples of IBM patients, but neither in other idiopathic inflammatory myopathies, nor in muscle dystrophies. This suggests a specific role in the pathology of IBM (Figure 5). The level of NLRP3 upregulation correlates to levels of inflammatory cytokines and to ubiquitin, a degradation marker for proteins. Moreover, an increased mRNA expression of NLRC5 in IBM muscle compared to the control could be observed (Figure 1B). Previous in vitro studies demonstrated an inducibility of NLRC5 by IFN-γ and the activating function of NLRC5 on genes encoding MHCI. Knockout trials further confirmed IFN-γ-induced MHCI upregulation predominantly through NLRC5 [25]. Upregulated in response to cytokines, components of the protein aggregates can activate the NLRP3 inflammasome as a part of the IBM pathology. The lack of a physiological suppressor of the inflammasome leads to an increased release of IL-1β, which itself contributes to the deposition of aggregated proteins [11]. Besides activation by β-amyloid, NLRP3 is also induced by dysfunctional autophagic activity and reactive oxygen species (ROS) [26,27], which have been described to play a pivotal role in IBM pathophysiology [28]. Of particular interest in this interplay is the autophagic machinery, which has been shown to be involved in accumulation of β-amyloid in muscle fibers [23,29] and regulation of MHC-I internalization [30].

As the entire biopsy material was used for the RNA extraction, in addition to muscle cells, connective and adipose tissue, endothelium, and infiltrative cells were present in the biopsy and used as a whole for RNA extraction. Therefore, our study is restricted to the description of the quantitative regulation, but no statement on the mRNA localization, distribution, and/or sites of synthesis can be made. A comparison of the in vitro experiments with ex vivo experiments leads to the conclusion that in myotubes, NLRP3 is regulated independently from infiltrative immune cells or fibroblasts upon pro-inflammatory conditions, as occurring in IBM pathology.

Follow-up studies are required to define the exact role of NLRP3 in the pathogenesis of IBM. In addition to an analysis of the regulation of the effectors and end products IL-1β, IL-18, ASC, and caspase-1 in correlation with the NLRP3, a further correlation of NLRP3 to protein aggregations and a histological representation of the distribution pattern of the NLRP3 are favored. Moreover, a future analysis of autophagy markers including LC3-II and p62 in conjunction with NLRP3 could yield important insight in IBM pathogenesis. Collectively, our results suggest that the inflammasome around NLRP3 is a crucial element in IBM pathology. 

## 4. Materials and Methods

### 4.1. Patients and Muscle Biopsies

Diagnostic biopsies were used from skeletal muscle of 68 patients with myositis and muscular dystrophy (Table 1). All IBM, PM, DM, NM and muscular dystrophy samples were taken from the collection of the Department of Neuropathology, University Medical Center Göttingen, Germany. The study was approved by the ethics committee of the University Medical Center Göttingen, Germany (approval code: 5/2/16; approval date: 19 July 2017). IRB-approved informed consent for the use of samples has been obtained from patients.

### 4.2. Cells

The rhabdomyosarcoma cell line CCL136 (also referred to as myoblasts) was obtained from American Type Culture Collection (ATCC, Manassas, VA, USA). Primary myotube cultures were obtained from human donors during trauma surgery and orthopedic surgery. The procedure was approved by the local ethics committee (IRB). CCL136 cells were maintained in DMEM (Thermo Fisher Scientific, Dreieich, Germany) supplemented with 10% FCS (PAA laboratories Inc. by Thermo Fisher Scientific), 1% penicillin/streptomycin (Biochrom GmbH, Berlin, Germany), and 1% L-glutamine (Thermo Fisher Scientific), and the adherent primary cell line was cultivated in Skeletal Muscle Growth Medium (Promocell GmbH, Heidelberg, Germany). Cell cultures were maintained in a humidified incubator at 37 °C and 5% CO_2_. Depending on chamber or well size, cells were seeded at densities from 1 × 10^4^ to 2.5 × 10^5^ per chamber/well in chamber slides or 6-well plates (Thermo Fisher Scientific). The cells were maintained in the chamber slides or well plates up to a confluence of 70–80% (for myoblasts) or complete differentiation of the primary cells into myotubes. To this end, primary cells were transferred into the differentiation medium (Promocell) to promote complete differentiation. Cell cultures were treated with IL-1β 20 ng/mL and IFN-γ 300 U/mL in serum-free x-vivo medium (Lonza Group, Basel, Switzerland) for 24 h, 48 h, and 72 h. 

### 4.3. Immunohistochemistry 

Sections of 5 μm frozen muscle biopsy specimens were fixed in 4% paraformaldehyde at −20 °C for 5 min. Non-specific binding was reduced by 30 min incubation with 5% bovine serum albumin (BSA) and 3% goat serum (all from Jackson ImmunoResearch, West Grove, PA, USA) in Tris buffered saline (TBS, 0.05M, pH 7.4, 0.15M saline). All primary and secondary reagents were diluted in 1% BSA in TBS. The primary anti-human NLRP3 antibody (Abcam, Cambridge, UK) was used at a concentration of 1:200 over night at 4 °C. Immunoreactivity was detected using Alexa-594-conjugated pre-adsorbed secondary goat antibody against rabbit (from Life Technologies by Thermo Fischer Scientific). Nuclear counterstaining was performed by 4’,6-diamidino-2-phenylindole (DAPI, Molecular probes/Invitrogen by Thermo Fischer Scientific) at 1:50,000 for 1 min, followed by mounting in Fluoromount G (Electron Microscopy Sciences, Hatfield, PA, USA). Immunofluorescent microscopy and digital photography were performed on a Zeiss Axiophot microscope (Zeiss, Göttingen, Germany) using appropriate filters for red (594 nm) and blue (350 nm) fluorescence and a cooled CCD digital camera (Retiga 1300, Qimaging, Burnaby, BC, Canada) using the Qcapture software.

### 4.4. Immunostaining

Fixation of the cells was performed by incubation in 4% PFA for 10 min at room temperature. Cells were subsequently washed for 5 min in PBS and incubated in ice-cold methanol for 10 min. Afterwards, cells were incubated in a 1:1 solution of 10% BSA/PBS and 100% goat serum for 1 h at room temperature. Primary antibodies (rat anti-MHCI, AbD Serotec, Bio-Rad Laboratories, Neuried, Germany; rabbit anti- NLRP3, Abcam) were diluted 1:200 in 1% BSA/PBS, and cells were incubated overnight at 4 °C in a humid chamber. On the following day, cells were washed three times for 5 min in PBS and incubated with the secondary antibody (goat anti-rat AF568, goat-anti-rabbit AF488, Invitrogen by Thermo Fisher Scientific) 1:500 in 1% BSA/PBS for 1 h at room temperature. Nucleus counterstain was performed using 4′,6-diamidine-2-phenylindole (DAPI) for 2 min at room temperature and cells were embedded using Fluoromount-G (Biozol, München, Germany). Image acquisition was performed using an inverted transmitted light microscope Axiovert 200M (Zeiss) and fluorescence intensity was measured using a greyscale analysis (Image J 1.46 by Wayne Rasband, National Institute of Health, Bethesda, MD, USA). 

### 4.5. Western Blot

For protein extraction, myotube cultures were kept and stimulated in 6-well plates. After the indicated time of cytokine treatment, cells were washed with PBS, followed by 200 µL RIPA buffer mixed with PhosSTOP (Roche by Thermo Fisher Scientific) for each well. All further steps were performed on ice. The cell lysates were centrifuged for 15 min at 4 °C and maximum speed. The supernatant was removed for later use and protein concentration was determined using the Bradford assay. Briefly, 50 µL of BSA protein standard were pipetted into a 96-well plate in ascending concentrations of 0 to 500 µg/mL. The protein samples (1 µL to 49 µL water) were applied. Finally, 150 µL of Bradford reagent (Bio-Rad Protein Assay 1:50,000, Bio-Rad) was added. Protein concentration was determined by measuring the extinction coefficient at λ = 595 nm.

Cell lysates were boiled in Lämmli buffer in a 1:6 dilution at 95 °C for 5 min prior loading onto a polyacrylamide gel, and proteins were separated at 100 volts for up to 2 h. After size separation, the proteins were transferred onto a methanol-activated polyvinylidene fluoride (PVDF) membrane in a blot chamber using a wet-blot process. Blotting was performed in a transfer buffer (250 nM Tris, 1.92 M Glycin, 20% methanol in H_2_O, pH 8.3) at 110 volts for 10–15 min using the iBlot 2 Dry Blotting system (Invitrogen by Thermo Fisher Scientific) according to the manufacturer’s instructions. Membranes were blocked in 5% BSA/TBS-Tween 0.1% for 1 h at room temperature and incubated with the primary antibody (rabbit anti-NLRP3, 1:1000, Abcam; mouse anti-β-actin, 1:5000, Sigma-Aldrich, St.Louis, MI, USA) diluted in the block solution overnight at 4 °C. After 5 washing steps with TBS-Tween each for 5 min, membranes were incubated with the secondary antibody (goat anti-mouse, goat anti-rabbit, 1:5000, Jackson ImmunoResearch) diluted in 5% BSA/TBS-Tween 0.1% for 1.5 h at room temperature. Membranes were subsequently washed with TBS-Tween and developed using Pierce ECL Western Blotting Substrate (Fischer Thermo Scientific). Images of the stained PVDF membranes were taken using a Fusion FX (Vilber Lourmat GmbH, Eberhardzell, Germany), displayed with the associated program Vision Capt (Analis, Suarlee, Belgium), and band intensity was measured using greyscale analysis (Image J 1.46 by Wayne Rasband, NIH).

### 4.6. RNA Extraction

Myotubes were maintained in 6-well plates and stimulated with cytokines, as described above. The medium was removed and 350 µL RLT lysis buffer mixed with 1% β-mercaptoethanol was added into each well and incubated for 30 min at room temperature. The cell lysates were centrifuged at maximum speed and 4 °C for 3 min. Ethanol (70%, 700 µL) was added to the supernatant. 

The tissue from patient biopsy material was homogenized in 350 µL trizole using bead tubes and the addition of chloroform. Subsequent steps for RNA extraction from cells and tissue were performed using the RNeasy Mini Kit (Qiagen, Hilden, Germany) according to the manufacturer’s instructions. Purified RNA was stored at −80 °C.

### 4.7. Quantitative (Realtime) PCR (qPCR)

The synthesis of complementary DNA from the previously isolated mRNA was performed using the SuperScript II Kit (Invitrogen by Thermo Fisher Scientific) and oligo-dt-primer. The qPCR was carried out using TaqMan probes (NLRP3 (Hs00918082_m), NLRC5 (Hs1072148_m), TGFβ (Hs9999918_m), TNFα (Hs00174128_m), Ubiquitin (Hs00430290_m), APP (Hs00169098_m)). TaqMan Gene Expression Assay (Applied Biosystems by Thermo Fisher Scientific) in the Quant Studio System (Applied Biosystems by Thermo Fischer Scientific). GAPDH (Hs99999905_m) was used as the endogenous control. Each sample was run as technical triplicate. The PCR program was as follows: initialization 50 °C for 2 min, heat inactivation 95 °C for 10 min, denaturation 95 °C for 25 s, for 40 cycles, and a final elongation step at 60 °C for 1 min. 

### 4.8. Statistical Analysis

Statistical analysis was performed using the Graph Pad Prism 6 program (GraphPad, San Diego, CA, USA). Column statistics were used to test the normal distribution of the values using the D’Agostino–Pearson omnibus normality test. Data sets with normal distribution were analyzed using the unpaired Student’s *t*-test. On the remaining data sets, the Mann–Whitney test was applied. Correlation analysis was done with the Pearson correlation test. * *p* < 0.05 and ** *p* < 0.01were used as significant values.

## Figures and Tables

**Figure 1 ijms-24-10675-f001:**
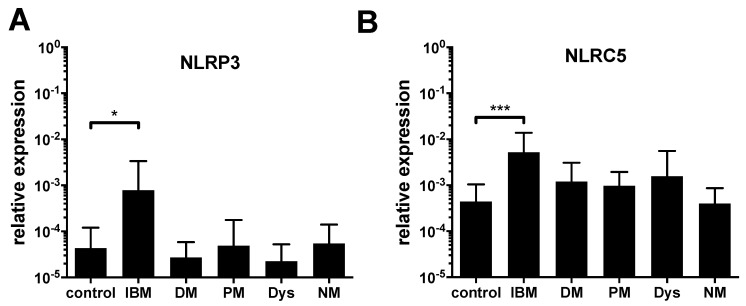
Expression analysis in muscle biopsy material from patients with inclusion body myositis shows an increased expression of NLRP3 and NLRC5. Quantification of the mRNA expression level of NLRP3 (**A**) and NLRC5 (**B**) relative to the endogenous control GAPDH in muscle biopsy material from healthy controls (n = 11), inclusion body myositis (IBM, n = 14), dermatomyositis (DM, n = 10), polymyositis (PM, n = 13), muscular dystrophy (Dys, n = 10), and necrotizing myopathy (NM, n = 10) patients. (ANOVA, levels of significance indicated by * *p* = 0.0492, *** *p* = 0.0006).

**Figure 2 ijms-24-10675-f002:**
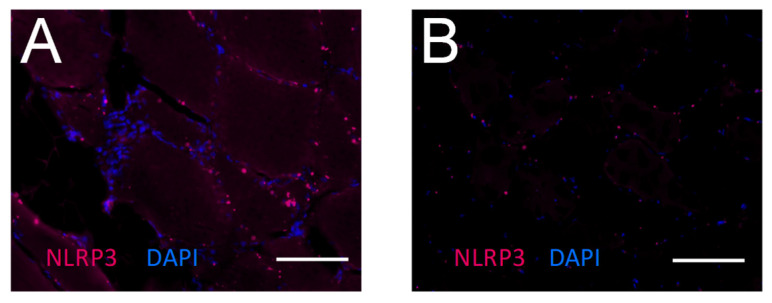
Immunohistochemical detection of NLRP3 in a muscle biopsy from a patient with inclusion body myositis (**A**) and a muscle biopsy of a non-myopathic control (**B**). Muscle tissue was immunostained for NLRP3 (red). Nuclei were counterstained using DAPI (blue). Image acquisition was performed using an inverted transmitted light microscope. Scale bar corresponds to 100 µm. An elevated expression level of NLRP3 is noted in many fibers from IBM muscle (**A**) in comparison to the control (**B**).

**Figure 3 ijms-24-10675-f003:**
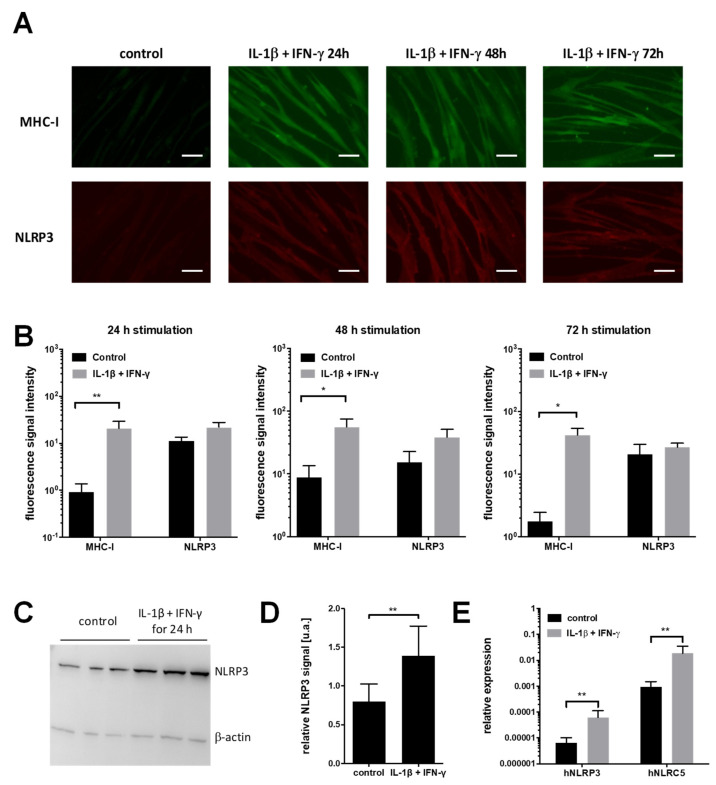
Stimulation with IL-1β and IFN-γ of muscle cells (myotubes) leads to an increased expression of MHC-I and NLRP3. (**A**) Immunfluorescence images of muscle cells without and with stimulation with IL-1β (20 ng/mL) and IFN-γ (300 U/mL) for 24 h, 48 h and 72 h, respectively. Cells were fixed at indicated times post treatment and immunostained for MHC-I (green) and NLRP3 (red). Image acquisition was performed using an inverted transmitted light microscope. Scale bar corresponds to 40 µm. (**B**) Quantification of signal intensity of MHC-I and NLRP3 without and with 24 h, 48 h, and 72 h stimulation, respectively. For each stimulated group, the signal increase for MHC-I was significant compared to the unstimulated control (Mann–Whitney Test; 24 h: n = 6, ** *p* = 0.0087; 48 h: n = 6, * *p* = 0.041; 72 h: n = 4, * *p* = 0.028). The NLRP3 signal intensity showed a tendency towards higher values upon stimulation without reaching statistical significance (mean signal intensity control vs. IL-1β and IFN-γ; 24 h: 11.1 vs. 21.7, *p* = 0.13; 48 h: 15.2 vs. 38.3, *p* = 0.065; 72 h: 20.9 vs. 26.8, *p* = 0.34). (**C**) Representative image of immunoblot membrane showing three untreated and three samples after 24 h stimulation with IL-1β and IFN-γ. An amount of 30 µg cell lysate was loaded per sample. NLRP3-specific bands are visible at approx. 116 kDa. β-actin (42 kDa) was used as loading control. (**D**) Quantification of western blot analysis shows a significant increase of NLRP3 expression (n = 6, Mann–Whitney Test, ** *p*= 0.0079) (**E**) Quantification of the mRNA expression levels of NLRP3 and NLRC5 relative to the endogenous control GAPDH. In comparison to the untreated control group, a significant upregulation on mRNA level after stimulation with IL-1β (20 ng/mL) and IFN-γ (300 U/mL) for 24 h was observed for NLRP3 and NLRC5, respectively. (n = 8, Student’s *t*-test). Graphs show mean with SD. Level of significance indicated by ** *p*<0.01.

**Figure 4 ijms-24-10675-f004:**
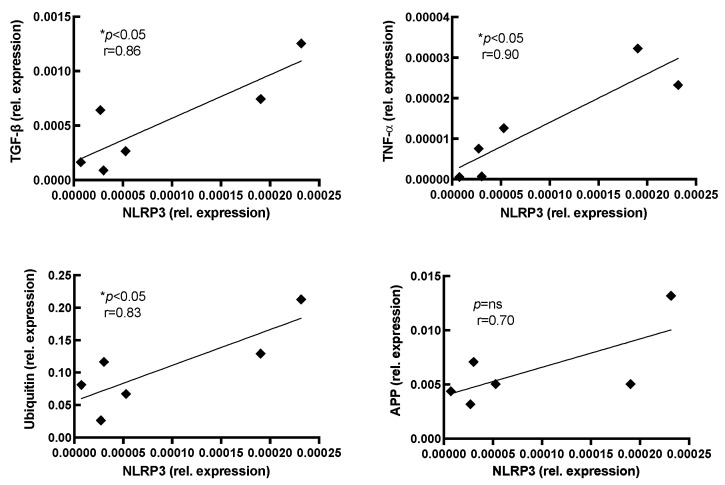
Correlation analysis of mRNA expression in IBM muscle. The mRNA expression of each marker from inclusion body myositis muscle samples was correlated to other pro-inflammatory and protein degradation markers. Each sample is represented by a diamond (◆). The mRNA expression of NLRP3 significantly correlates with that of TNF-α, TGF-β, and ubiquitin. The expression of amyloid precursor protein (APP) weakly correlates with NLRP3 expression levels and does not reach statistical significance (n = 6; Pearson correlation). Level of significance indicated by * *p* < 0.05.

**Figure 5 ijms-24-10675-f005:**
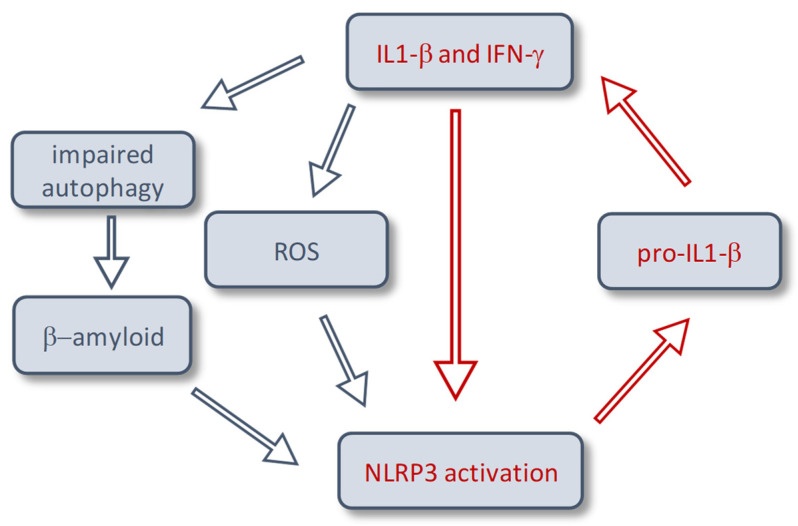
Schematic diagram depicting the proposed mechanism of NLRP3 activation in inclusion body myositis. Chronic inflammatory stimulus with IL-1β and IFN-γ results in impaired autophagy, reactive oxygen species and accumulation of β-amyloid. This leads to an activation of the NLRP3 inflammasome and subsequently higher levels of IL-1β, and a subsequent vicious cycle leading to more deposition of β-amyloid. Blue arrows indicate previously published pathways. Red arrows depict mechanisms proposed on the basis of this publication.

**Table 1 ijms-24-10675-t001:** Overview of patient samples indicating disease, sex, and mean age at time of biopsy.

	n	Male (%)	Female (%)	Average Age at Biopsy
Total	68	56	44	54
Control	11	36	64	48
Inclusion body myositis	14	79	21	64
Dermatomyositis	10	50	50	54
Polymyositis	13	31	69	54
Necrotizing myopathy	10	70	30	55
Muscular dystrophy	10	70	30	47

## Data Availability

None of the data are available in a repository or other accessible database. The data will be made available to researchers upon request.

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
