# Peer review of "Inflammasome in Skeletal Muscle: NLRP3 Is an Inflammatory Cell Stress Component in Inclusion Body Myositis"

_ijms, 2023, doi:10.3390/ijms241310675_

Round 1

Reviewer 1 Report

According to the authors, the main objective of this study was to demonstrate if the inflammasome NLRP3 is a component in the inclusion body myositis (IBM), particularly by assessing both in vitro (culture cells) and in vivo (biopsies) assays. Even though interesting data were presented there are another couple of factors that should be concerned.

In the Abstract section:

1) It is important to present the p-value for the results with statistically significant differences.

In the Introduction section:

2) Please add some references that are missed in this section, e.g. in the last sentence of the first paragraph (lines 36-37) and in the second paragraph (lines 38-39).

3) Based on the pieces of information presented in the third paragraph, it is mandatory to clarify if the simple enhancement of MHC class I is directly associated with the increase in the activation of TCD8+ lymphocytes. In addition, is the increase in MHC class I associated with the increased presentation of self-peptides by muscle fibers, which could explain the clonal expansion of this immune cell?

4) It is well-known that NLPR3 is closely associated with the induction of pro-inflammatory cytokines, mainly IL-1beta and IL-18. Hence, why the authors neither mentioned nor evaluated   IL-18 in this study.

In the Results section

5) Please check the description of the "n" in the Figure 1 legend (line 89).

6) In Lines 118-119, the authors declared that "There was a tendency for signal increase for NLRP3 upon stimulation compared to the untreated control which showed no statistical significance Figure 3B)." However, where these data were presented in order to support this "tendency"? What were the p-values? Please check this sentence and present the values to support this suggestion.

7) In relation to the data presented in Figure 4 (correlation analysis), why the authors did not assess the main cytokines elicited by NLRP3, such as IL-1beta and IL-18? In addition, why the authors presented results of only 6 samples and not of the 14 samples obtained from individuals with IBM?

8) Why did the authors not measure the cytokines levels (protein) in the muscle biopsies?

In the Discussion section:

9) What means VCP? (line 184)

10) Please add more pieces of information concerning the role of IL-beta and IFN-gamma in the study context. What was the importance to use these cytokines and not others?

In the Material and Methods section:

11) Please add the Ethic Committee approval number.

12) Please present what parameters were parametric and non-parametric. In addition, what was set the p-value (>0.05)?

 Minor editing of English language required.

Author Response

Response to reviewer 1:

According to the authors, the main objective of this study was to demonstrate if the inflammasome NLRP3 is a component in the inclusion body myositis (IBM), particularly by assessing both in vitro (culture cells) and in vivo (biopsies) assays. Eventhough interesting data were presented, there are another couple of factors that should be concerned.

In the Abstract section:

Comment 1) It is important to present the p-value for the results with statistically significant differences.

            REPLY: We are grateful for this suggestion and have now added the p-values in the Abstract section with mentioning the statistically significant differences (page 1).

In the Introduction section:

Comment 2) Please add some references that are missed in this section, e.g. in the last sentence of the first paragraph (lines 36-37) and in the second paragraph (lines 38-39).

            REPLY: Thank you for pointing this out. The missing references have now been inserted (page 2).

Comment 3) Based on the pieces of information presented in the third paragraph, it is mandatory to clarify if the simple enhancement of MHC class I is directly associated with the increase in the activation of TCD8+ lymphocytes. In addition, is the increase in MHC class I associated with the increased presentation of self-peptides by muscle fibers, which could explain the clonal expansion of this immune cell?

            REPLY: We are happy to provide more details on this topic, which has been specifically addressed by our group as well as several others. There is indeed convincing evidence available that there is a tight association between MHC class I overexpression and activation of CD8+ T cells in IBM muscle. We have now extended the introduction section to make this point more clear (pages 1 and 2).

Comment 4) It is well-known that NLPR3 is closely associated with the induction of pro-inflammatory cytokines, mainly IL-1beta and IL-18. Hence, why the authors neither mentioned nor evaluated   IL-18 in this study.

            REPLY: We are thankful for this suggestion. To the best of our knowledge, the role of IL-18 in the pathogenesis of IBM is poorly studied so far. By contrast, there is clear evidence that IL-1beta does play a key role in IBM pathogenesis. This is the main reason for focusing on that cytokine rather than on IL-18. In light of our present results, we do agree with the reviewer that, in future studies, it will be of interest to evaluate the expression level of IL-18 in IBM.

We have now added information on IL-18 in the introduction part and in the discussion (pages 2 and 9).

In the Results section

Comment 5)

Please check the description of the "n" in the Figure 1 legend (line 89).

            REPLY: The “n” is now mentioned for every subgroup.

Comment 6) In Lines 118-119, the authors declared that "There was a tendency for signal increase for NLRP3 upon stimulation compared to the untreated control which showed no statistical significance Figure 3B)." However, where these data were presented in order to support this "tendency"? What were the p-values? Please check this sentence and present the values to support this suggestion.

            REPLY: By visual immunofluorescence analysis, our data demonstrate that signal intensity was higher in stimulated cells than in controls (see Figure 3A). However, by grey scale analysis, this observation could not be supported at a statistically significant level. The signal intensity values were higher in stimulated cells compared to controls as shown in Figure 3B.

We have now added p-values and mean values in the Figure legend to clarify our statement and interpretation.

Comment 7) In relation to the data presented in Figure 4 (correlation analysis), why the authors did not assess the main cytokines elicited by NLRP3, such as IL-1beta and IL-18? In addition, why the authors presented results of only 6 samples and not of the 14 samples obtained from individuals with IBM?

            REPLY: As stated above, IL-1beta has been studied extensively before, including a detailed and broad correlation analysis between IL-1beta and all relevant pro-inflammatory chemokines and cytokines as well as cell stress molecules and amyloid-associated parameters (Schmidt et al. Brain 2008; Schmidt et al. Brain 2012 etc. –see details of citations and full list of publications below). The reason for not studying IL-18 in the present study has been mentioned above in reply to comment #4. The number of six samples per parameter is explained by two major factors: a) the amount of RNA that was available from each subject was small so that it was important to use as little numbers of samples as possible; secondly (b), for financial restrictions, it was necessary to use as little molecular biology consumables as possible; thus, only the screening experiments for NLRP-3 could be performed with the whole set of subjects. Additional experiments such as the correlation experiment had to be performed with the smallest group size that would allow statistically valid results.

Comment 8) Why did the authors not measure the cytokines levels (protein) in the muscle biopsies?

            REPLY: We appreciate this comment. Quantitative analysis by Western blot or proteomics requires sufficient amount of tissue and sufficient financial resources. At the same time, protein quantification by Western would not be able to differentiate the location and cellular source of the cytokine. Future functional studies in myotubes and myoblasts will address these issues and evaluate the quantitative protein levels in the skeletal muscle as well as in the supernatant (=secreted molecules). The focus of the present work was more on the activation of NLRP3 itself as a result of increased muscle inflammation.

Comment 9) What means VCP? (line 184)

            REPLY: VCP stands for Valosin-Containing Protein. The abbreviation is now explained (page 8).

In the Discussion section:

Comment 10) Please add more pieces of information concerning the role of IL-beta and IFN-gamma in the study context. What was the importance to use these cytokines and not others?

            REPLY: In cell culture, IL-1beta and IFN-gamma can effectively mimic the pathological changes present in IBM (Schmidt et al., Brain 2008, Schmidt et al., Brain 2012 etc., see detailed and full publication list below). With this specific cytokine combination, the muscle cell responses to pro-inflammatory stress signals in IBM muscle samples and cell culture have been studied extensively and in a well reproducible fashion. As example, using this model system for evaluation of IBM pathology, the pathways of autophagy (Lünemann et al., Ann Neurol 2007; Keller et al., JBC 2011), NO-stress (Schmidt et al., Brain 2012), β-amyloid-accumulation (Schmidt et al., Mediators Inflamm 2017; Keller et al., J Neurol Sci 2013), HMGB1/RAGE stress signaling (Muth et al., Exp Neurol 2015), αB-crystallin as cell stress component in skeletal muscle cell stress (Muth et al., JNNP 2009) have been specifically studied before by our group.

These data are now explained in more detail in the discussion (page 8).

Comment 11) Please add the Ethic Committee approval number.

            REPLY: We have now inserted the Ethic Committee approval number in the methods section on page 9: (Ethic committee of the University Medical Center Göttingen, Germany; approval code: 5/2/16; approval date: 19 Jul 2017).

Comment 12) Please present what parameters were parametric and non-parametric. In addition, what was set the p-value (>0.05)?

            REPLY: For in-vitro expression studies, immunoflourescence and Western blot analysis, the non-parametric Mann-Whitney U test was used for comparison of two groups. For expression analysis from muscle biopsies, the parametric tests T-test and Pearson correlation were used when comparing two groups and ANOVA when comparing multiple groups. All tests are indicated in the corresponding figure legends.

The levels for p-values (*p<0.05 and **p<0.01) have now been added in Methods section (page 11).

Reviewer 2 Report

The article is very interesting, and clearly shows the presence of NLRP3 inflammasome in Inclusion body myositis (IBM), and hints at its potential aggravating role in the progression of the disease. Manuscript is very well written, results are well presented, and the figures are easy to understand. 

Major remarks:

1- Some experiments are needed to further address the role of NLRP3 in IBM. For instance, Authors present the proposed mechanism of NLRP3 in the schematic diagram found in figure 5, and discuss about the role of IL-1β, which is an end product of the NLRP3 inflammasome. Authors should test the mRNA expression (and or protein levels), of several NLRP3 effectors and end products, such as ASC, caspase-1, gasdermin D, IL-1β, IL-18. If the entire biopsy material was used for the RNA extraction, perhaps additional qPCR can help detect levels of IL-1β and other actors, or it can be simply done in the CCL136 cell line.

2- In addition, since autophagy is key player in the accumulation of β amyloid and a target of NLRP3- IL-1β, some markers/actors of autophagy can also be tested, such as LC3II/LC3I, p62, ULK1, Beclin-1, …

Minor remarks/Suggestions:

- Do authors have any data (or could test) whether inhibition of NLRP3 by specific inflammasome inhbitors (like MCC950, β-OH, OLT1177, MNS, ...) could have a positive impact on IBM; in this case reduction in IL-1β expression and production/secretion, and enhanced autophagy/less β amyloid accumulation.

Author Response

Response to Reviewer 2:

The article is very interesting, and clearly shows the presence of NLRP3 inflammasome in Inclusion body myositis (IBM), and hints at its potential aggravating role in the progression of the disease. Manuscript is very well written, results are well presented, and the figures are easy to understand.

Major remarks:

Comment 1): Some experiments are needed to further address the role of NLRP3 in IBM. For instance, Authors present the proposed mechanism of NLRP3 in the schematic diagram found in figure 5, and discuss about the role of IL-1β, which is an end product of the NLRP3 inflammasome. Authors should test the mRNA expression (and or protein levels), of several NLRP3 effectors and end products, such as ASC, caspase-1, gasdermin D, IL-1β, IL-18. If the entire biopsy material was used for the RNA extraction, perhaps additional qPCR can help detect levels of IL-1β and other actors, or it can be simply done in the CCL136 cell line.

            REPLY: We are grateful for the positive comment and excellent suggestions. The pro-inflammatory cytokines and chemokines as well as their correlation with IL-1beta have been studied extensively before: Schmidt et al., Brain 2008; Schmidt et al., Brain 2012 (see full reference list at the bottom of this reply letter). Further analysis of NLRP3 inflammasome-associated mediators in IBM that have not previously studied, such as IL-18, should be addressed in subsequent studies. The present report was focused on identifying the presence and role of NLRP3 in various myositis conditions and under pro-inflammatory functional conditions. We have now mentioned our previous work in more detail (page 8) and provide a more detailed outlook on future experiments (page 9).

Comment 2: In addition, since autophagy is key player in the accumulation of β amyloid and a target of NLRP3- IL-1β, some markers/actors of autophagy can also be tested, such as LC3II/LC3I, p62, ULK1, Beclin-1, …

Do authors have any data (or could test) whether inhibition of NLRP3 by specific inflammasome inhbitors (like MCC950, β-OH, OLT1177, MNS, ...) could have a positive impact on IBM; in this case reduction in IL-1β expression and production/secretion, and enhanced autophagy/less β amyloid accumulation.

            REPLY: Thank you for these excellent suggestions. The role of autophagy and protein accumulation has been extensively addressed in IBM muscle as well as in our pro-inflammatory cytokine model (Lünemann et al., AnnNeurol 2007; Keller et al. JBC 2011; Keller et al. J Neurol Sci 2013; Schmidt et al., Med Inflamm 2017; see full reference list at the bottom of this reply letter). Additional blocking experiments that functionally explore the relevance of the NLRP-3 inflammasome and pathways like autophagy or nitric oxide cell stress components will be specifically addressed in future experiments. At the same time, additional co-localization studies and quantitative protein expression studies will be studied in IBM muscle. Such experiments are beyond the scope of the present study.

We now provide additional aspects in the Discussion section (page 9) and had already illustrated these mechanisms -including autophagy- in Figure 5.

Previous literature from our group on mechanisms in myositis / IBM

Schmidt K, Wienken M, Keller CW, Balcarek P, Münz C, Schmidt J. IL-1β-Induced Accumulation of Amyloid: Macroautophagy in Skeletal Muscle Depends on ERK. Mediators Inflamm. 2017; 2017: 5470831.

Muth IE, Zschüntzsch J, Kleinschnitz K, Wrede A, Gerhardt E, Balcarek P, Schreiber-Katz O, Zierz S, Dalakas MC, Voll RE, Schmidt J. HMGB1 and RAGE in skeletal muscle inflammation: implications for protein accumulation in inclusion body myositis. Exp Neurol 2015; 271: 189–197.

Keller CW, Schmitz M, Münz C, Lünemann JD, Schmidt J. TNF-α upregulates macroautophagic processing of APP/β-amyloid in a human rhabdomyosarcoma cell line. J Neurol Sci 2013; 325: 103-7.

Zschüntzsch J, Voss J, Creus K, Sehmisch S, Raju R, Dalakas MC, Schmidt J. Provision of an explanation for the inefficacy of immunotherapy in sIBM: Quantitative assessment of inflammation and β-amyloid in the muscle. Arthritis Rheum 2012; 64: 4094-103.

Schmidt J, Barthel K, Zschüntzsch J, Muth IE, Swindle EJ, Hombach A, Sehmisch S, Wrede A, Lühder F, Gold R, Dalakas MC. Nitric oxide stress in sIBM muscle fibres: inhibition of iNOS prevents IL-1β-induced accumulation of β-amyloid and cell death. Brain 2012; 135: 1102-14.

Keller CW, Fokken C, Turville SG, Lünemann A, Schmidt J*, Münz C*, Lünemann JD*. TNF-α induces macroautophagy and regulates MHC Class II expression in human skeletal muscle cells. J Biol Chem 2011; 286: 3970-3980 *equal contribution.

Muth IE, Barthel K, Bähr M, Dalakas MC, Schmidt J. Pro-inflammatory cell stress in sIBM muscle: overexpression of αB-crystallin is associated with amyloid precursor protein and accumulation of β-amyloid. J Neurol Neurosurg Psychiatry 2009; 80: 1344-9.

Schmidt J, Barthel K, Wrede A, Salajegheh M, Bähr M, Dalakas MC. Interrelation of inflammation and APP in sIBM: IL-1beta induces accumulation of beta-amyloid in skeletal muscle. Brain 2008; 131: 1228-40.

Lunemann JD*, Schmidt J*, Schmid D, Barthel K, Wrede A, Dalakas MC, Münz C. β-amyloid is a Substrate of Macroautophagy in Sporadic Inclusion Body Myositis. Ann Neurol 2007; 61: 476-483; *equal contribution.

Salajegheh M, Raju R, Schmidt J, Dalakas MC. Upregulation of Thrombospondin-1 (TSP-1) and its binding partners, CD36 and CD47, in sporadic inclusion body myositis. J Neuroimmunol 2007; 187: 166-74.

Schmidt J, Rakocevic G, Raju R, Dalakas MC. Upregulated inducible costimulator and ICOS-L in IBM muscle: significance for CD8+ T cell cytotoxicity. Brain 2004; 127:1182-90.

Round 2

Reviewer 2 Report

The authors have addressed the concerns raised in the first round of review, provided the required data and made the necessary changes to the text. Overall, this is an interesting and well-written article.